# WORD TRANSLATION WITHOUT PARALLEL DATA

**Guillaume Lample**[*][†][‡]**, Alexis Conneau**[*][†][§]**,**
**Marc'Aurelio Ranzato**[†]**, Ludovic Denoyer**[‡]**, Hervé Jégou**[†]
{glample,aconneau,ranzato,rvj}@fb.com
ludovic.denoyer@upmc.fr

## ABSTRACT

State-of-the-art methods for learning cross-lingual word embeddings have relied on bilingual dictionaries or parallel corpora. Recent studies showed that the need for parallel data supervision can be alleviated with character-level information. While these methods showed encouraging results, they are not on par with their supervised counterparts and are limited to pairs of languages sharing a common alphabet. In this work, we show that we can build a bilingual dictionary between two languages without using any parallel corpora, by aligning monolingual word embedding spaces in an unsupervised way. Without using any character information, our model even outperforms existing supervised methods on cross-lingual tasks for some language pairs. Our experiments demonstrate that our method works very well also for distant language pairs, like English-Russian or English-Chinese. We finally describe experiments on the English-Esperanto low-resource language pair, on which there only exists a limited amount of parallel data, to show the potential impact of our method in fully unsupervised machine translation. Our code, embeddings and dictionaries are publicly available[1].

## 1 INTRODUCTION

Most successful methods for learning distributed representations of words (e.g. Mikolov et al. (2013c;a); Pennington et al. (2014); Bojanowski et al. (2017)) rely on the distributional hypothesis of Harris (1954), which states that words occurring in similar contexts tend to have similar meanings. Levy & Goldberg (2014) show that the skip-gram with negative sampling method of Mikolov et al. (2013c) amounts to factorizing a word-context co-occurrence matrix, whose entries are the pointwise mutual information of the respective word and context pairs. Exploiting word co-occurrence statistics leads to word vectors that reflect the semantic similarities and dissimilarities: similar words are close in the embedding space and conversely.

Mikolov et al. (2013b) first noticed that continuous word embedding spaces exhibit similar structures across languages, even when considering distant language pairs like English and Vietnamese. They proposed to exploit this similarity by learning a linear mapping from a source to a target embedding space. They employed a parallel vocabulary of five thousand words as anchor points to learn this mapping and evaluated their approach on a word translation task. Since then, several studies aimed at improving these cross-lingual word embeddings (Faruqui & Dyer (2014); Xing et al. (2015); Lazaridou et al. (2015); Ammar et al. (2016); Artetxe et al. (2016); Smith et al. (2017)), but they all rely on bilingual word lexicons.

Recent attempts at reducing the need for bilingual supervision (Smith et al., 2017) employ identical character strings to form a parallel vocabulary. The iterative method of Artetxe et al. (2017) gradually aligns embedding spaces, starting from a parallel vocabulary of aligned digits. These methods are however limited to similar languages sharing a common alphabet, such as European languages. Some recent methods explored distribution-based approach (Cao et al., 2016) or adversarial training Zhang et al. (2017b) to obtain cross-lingual word embeddings without any parallel data. While these

---

[*]Equal contribution. Order has been determined with a coin flip.

[†]Facebook AI Research

[‡]Sorbonne Universités, UPMC Univ Paris 06, UMR 7606, LIP6

[§]LIUM, University of Le Mans

[1]https://github.com/facebookresearch/MUSE

approaches sound appealing, their performance is significantly below supervised methods. To sum up, current methods have either not reached competitive performance, or they still require parallel data, such as aligned corpora (Gouws et al., 2015; Vulic & Moens, 2015) or a seed parallel lexicon (Duong et al., 2016).

In this paper, we introduce a model that either is on par, or outperforms supervised state-of-the-art methods, without employing any cross-lingual annotated data. We only use two large monolingual corpora, one in the source and one in the target language. Our method leverages adversarial training to learn a linear mapping from a source to a target space and operates in two steps. First, in a two-player game, a discriminator is trained to distinguish between the mapped source embeddings and the target embeddings, while the mapping (which can be seen as a generator) is jointly trained to fool the discriminator. Second, we extract a synthetic dictionary from the resulting shared embedding space and fine-tune the mapping with the closed-form Procrustes solution from Schönemann (1966). Since the method is unsupervised, cross-lingual data can not be used to select the best model. To overcome this issue, we introduce an unsupervised selection metric that is highly correlated with the mapping quality and that we use both as a stopping criterion and to select the best hyper-parameters.

In summary, this paper makes the following main contributions:

- We present an unsupervised approach that reaches or outperforms state-of-the-art supervised approaches on several language pairs and on three different evaluation tasks, namely word translation, sentence translation retrieval, and cross-lingual word similarity. On a standard word translation retrieval benchmark, using 200k vocabularies, our method reaches 66.2% accuracy on English-Italian while the best supervised approach is at 63.7%.

- We introduce a cross-domain similarity adaptation to mitigate the so-called hubness problem (points tending to be nearest neighbors of many points in high-dimensional spaces). It is inspired by the self-tuning method from Zelnik-manor & Perona (2005), but adapted to our two-domain scenario in which we must consider a bi-partite graph for neighbors. This approach significantly improves the absolute performance, and outperforms the state of the art both in supervised and unsupervised setups on word-translation benchmarks.

- We propose an unsupervised criterion that is highly correlated with the quality of the mapping, that can be used both as a stopping criterion and to select the best hyper-parameters.

- We release high-quality dictionaries for 12 oriented languages pairs, as well as the corresponding supervised and unsupervised word embeddings.

- We demonstrate the effectiveness of our method using an example of a low-resource language pair where parallel corpora are not available (English-Esperanto) for which our method is particularly suited.

The paper is organized as follows. Section 2 describes our unsupervised approach with adversarial training and our refinement procedure. We then present our training procedure with unsupervised model selection in Section 3. We report in Section 4 our results on several cross-lingual tasks for several language pairs and compare our approach to supervised methods. Finally, we explain how our approach differs from recent related work on learning cross-lingual word embeddings.

## 2 MODEL

In this paper, we always assume that we have two sets of embeddings trained independently on monolingual data. Our work focuses on learning a mapping between the two sets such that translations are close in the shared space. Mikolov et al. (2013b) show that they can exploit the similarities of monolingual embedding spaces to learn such a mapping. For this purpose, they use a known dictionary of $n = 5000$ pairs of words $\{x_i, y_i\}_{i \in \{1, n\}}$, and learn a linear mapping $W$ between the source and the target space such that

$$W^\star = \underset{W \in M_d(\mathbb{R})}{\operatorname{argmin}} \|WX - Y\|_{\mathrm{F}} \qquad (1)$$

where $d$ is the dimension of the embeddings, $M_d(\mathbb{R})$ is the space of $d \times d$ matrices of real numbers, and $X$ and $Y$ are two aligned matrices of size $d \times n$ containing the embeddings of the words in the parallel vocabulary. The translation $t$ of any source word $s$ is defined as $t = \operatorname{argmax}_t \cos(Wx_s, y_t)$.

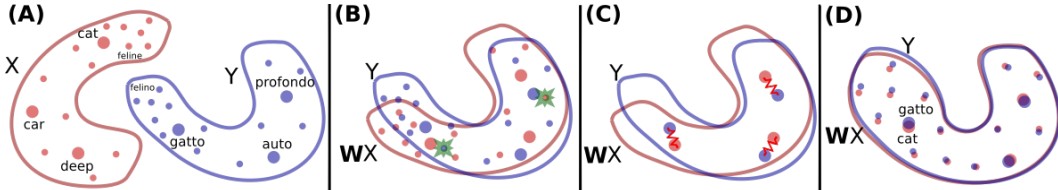

**Figure 1: Toy illustration of the method.** **(A)** There are two distributions of word embeddings, English words in red denoted by $X$ and Italian words in blue denoted by $Y$, which we want to align/translate. Each dot represents a word in that space. The size of the dot is proportional to the frequency of the words in the training corpus of that language. **(B)** Using adversarial learning, we learn a rotation matrix $W$ which roughly aligns the two distributions. The green stars are randomly selected words that are fed to the discriminator to determine whether the two word embeddings come from the same distribution. **(C)** The mapping $W$ is further refined via Procrustes. This method uses frequent words aligned by the previous step as anchor points, and minimizes an energy function that corresponds to a spring system between anchor points. The refined mapping is then used to map all words in the dictionary. **(D)** Finally, we translate by using the mapping $W$ and a distance metric, dubbed CSLS, that expands the space where there is high density of points (like the area around the word "cat"), so that "hubs" (like the word "cat") become less close to other word vectors than they would otherwise (compare to the same region in panel (A)).

In practice, Mikolov et al. (2013b) obtained better results on the word translation task using a simple linear mapping, and did not observe any improvement when using more advanced strategies like multilayer neural networks. Xing et al. (2015) showed that these results are improved by enforcing an orthogonality constraint on $W$. In that case, the equation (1) boils down to the Procrustes problem, which advantageously offers a closed form solution obtained from the singular value decomposition (SVD) of $YX^T$:

$$W^\star = \underset{W \in O_d(\mathbb{R})}{\operatorname{argmin}} \|WX - Y\|_{\mathrm{F}} = UV^T, \text{with } U\Sigma V^T = \operatorname{SVD}(YX^T). \tag{2}$$

In this paper, we show how to learn this mapping $W$ without cross-lingual supervision; an illustration of the approach is given in Fig. 1. First, we learn an initial proxy of $W$ by using an adversarial criterion. Then, we use the words that match the best as anchor points for Procrustes. Finally, we improve performance over less frequent words by changing the metric of the space, which leads to spread more of those points in dense regions. Next, we describe the details of each of these steps.

## 2.1 Domain-adversarial setting

In this section, we present our domain-adversarial approach for learning $W$ without cross-lingual supervision. Let $\mathcal{X} = \{x_1, ..., x_n\}$ and $\mathcal{Y} = \{y_1, ..., y_m\}$ be two sets of $n$ and $m$ word embeddings coming from a source and a target language respectively. A model is trained to discriminate between elements randomly sampled from $W\mathcal{X} = \{Wx_1, ..., Wx_n\}$ and $\mathcal{Y}$. We call this model the discriminator. $W$ is trained to prevent the discriminator from making accurate predictions. As a result, this is a two-player game, where the discriminator aims at maximizing its ability to identify the origin of an embedding, and $W$ aims at preventing the discriminator from doing so by making $W\mathcal{X}$ and $\mathcal{Y}$ as *similar* as possible. This approach is in line with the work of Ganin et al. (2016), who proposed to learn latent representations invariant to the input domain, where in our case, a domain is represented by a language (source or target).

**Discriminator objective** We refer to the discriminator parameters as $\theta_D$. We consider the probability $P_{\theta_D}(\text{source} = 1 | z)$ that a vector $z$ is the mapping of a source embedding (as opposed to a target embedding) according to the discriminator. The discriminator loss can be written as:

$$\mathcal{L}_D(\theta_D | W) = -\frac{1}{n}\sum_{i=1}^{n} \log P_{\theta_D}(\text{source} = 1 | Wx_i) - \frac{1}{m}\sum_{i=1}^{m} \log P_{\theta_D}(\text{source} = 0 | y_i). \tag{3}$$

**Mapping objective** In the unsupervised setting, $W$ is now trained so that the discriminator is unable to accurately predict the embedding origins:

$$\mathcal{L}_W(W | \theta_D) = -\frac{1}{n}\sum_{i=1}^{n} \log P_{\theta_D}(\text{source} = 0 | Wx_i) - \frac{1}{m}\sum_{i=1}^{m} \log P_{\theta_D}(\text{source} = 1 | y_i). \tag{4}$$

**Learning algorithm** To train our model, we follow the standard training procedure of deep adversarial networks of Goodfellow et al. (2014). For every input sample, the discriminator and the mapping matrix $W$ are trained successively with stochastic gradient updates to respectively minimize $\mathcal{L}_D$ and $\mathcal{L}_W$. The details of training are given in the next section.

## 2.2 REFINEMENT PROCEDURE

The matrix $W$ obtained with adversarial training gives good performance (see Table 1), but the results are still not on par with the supervised approach. In fact, the adversarial approach tries to align all words irrespective of their frequencies. However, rare words have embeddings that are less updated and are more likely to appear in different contexts in each corpus, which makes them harder to align. Under the assumption that the mapping is linear, it is then better to infer the global mapping using only the most frequent words as anchors. Besides, the accuracy on the most frequent word pairs is high after adversarial training.

To refine our mapping, we build a synthetic parallel vocabulary using the $W$ just learned with adversarial training. Specifically, we consider the most frequent words and retain only mutual nearest neighbors to ensure a high-quality dictionary. Subsequently, we apply the Procrustes solution in (2) on this generated dictionary. Considering the improved solution generated with the Procrustes algorithm, it is possible to generate a more accurate dictionary and apply this method iteratively, similarly to Artetxe et al. (2017). However, given that the synthetic dictionary obtained using adversarial training is already strong, we only observe small improvements when doing more than one iteration, i.e., the improvements on the word translation task are usually below $1\%$.

## 2.3 CROSS-DOMAIN SIMILARITY LOCAL SCALING (CSLS)

In this subsection, our motivation is to produce reliable matching pairs between two languages: we want to improve the comparison metric such that the nearest neighbor of a source word, in the target language, is more likely to have as a nearest neighbor this particular source word.

Nearest neighbors are by nature asymmetric: $y$ being a $K$-NN of $x$ does not imply that $x$ is a $K$-NN of $y$. In high-dimensional spaces (Radovanović et al., 2010), this leads to a phenomenon that is detrimental to matching pairs based on a nearest neighbor rule: some vectors, dubbed *hubs*, are with high probability nearest neighbors of many other points, while others (anti-hubs) are not nearest neighbors of any point. This problem has been observed in different areas, from matching image features in vision (Jegou et al., 2010) to translating words in text understanding applications (Dinu et al., 2015). Various solutions have been proposed to mitigate this issue, some being reminiscent of pre-processing already existing in spectral clustering algorithms (Zelnik-manor & Perona, 2005).

However, most studies aiming at mitigating hubness consider a single feature distribution. In our case, we have two domains, one for each language. This particular case is taken into account by Dinu et al. (2015), who propose a pairing rule based on reverse ranks, and the inverted soft-max (ISF) by Smith et al. (2017), which we evaluate in our experimental section. These methods are not fully satisfactory because the similarity updates are different for the words of the source and target languages. Additionally, ISF requires to cross-validate a parameter, whose estimation is noisy in an unsupervised setting where we do not have a direct cross-validation criterion.

In contrast, we consider a bi-partite neighborhood graph, in which each word of a given dictionary is connected to its $K$ nearest neighbors in the other language. We denote by $\mathcal{N}_{\mathrm{T}}(Wx_s)$ the neighborhood, on this bi-partite graph, associated with a mapped source word embedding $Wx_s$. All $K$ elements of $\mathcal{N}_{\mathrm{T}}(Wx_s)$ are words from the target language. Similarly we denote by $\mathcal{N}_{\mathrm{S}}(y_t)$ the neighborhood associated with a word $t$ of the target language. We consider the mean similarity of a source embedding $x_s$ to its target neighborhood as

$$r_{\mathrm{T}}(Wx_s) = \frac{1}{K} \sum_{y_t \in \mathcal{N}_{\mathrm{T}}(Wx_s)} \cos(Wx_s, y_t), \qquad (5)$$

where $\cos(.,.)$ is the cosine similarity. Likewise we denote by $r_{\mathrm{S}}(y_t)$ the mean similarity of a target word $y_t$ to its neighborhood. These quantities are computed for all source and target word vectors with the efficient nearest neighbors implementation by Johnson et al. (2017). We use them to define a similarity measure $\mathrm{CSLS}(.,.)$ between mapped source words and target words, as

$$\mathrm{CSLS}(Wx_s, y_t) = 2\cos(Wx_s, y_t) - r_{\mathrm{T}}(Wx_s) - r_{\mathrm{S}}(y_t). \qquad (6)$$

Intuitively, this update increases the similarity associated with isolated word vectors. Conversely it decreases the ones of vectors lying in dense areas. Our experiments show that the CSLS significantly increases the accuracy for word translation retrieval, while not requiring any parameter tuning.

## 3 TRAINING AND ARCHITECTURAL CHOICES

### 3.1 ARCHITECTURE

We use unsupervised word vectors that were trained using fastText[2]. These correspond to monolingual embeddings of dimension 300 trained on Wikipedia corpora; therefore, the mapping $W$ has size $300 \times 300$. Words are lower-cased, and those that appear less than 5 times are discarded for training. As a post-processing step, we only select the first 200k most frequent words in our experiments.

For our discriminator, we use a multilayer perceptron with two hidden layers of size 2048, and Leaky-ReLU activation functions. The input to the discriminator is corrupted with dropout noise with a rate of 0.1. As suggested by Goodfellow (2016), we include a smoothing coefficient $s = 0.2$ in the discriminator predictions. We use stochastic gradient descent with a batch size of 32, a learning rate of 0.1 and a decay of 0.95 both for the discriminator and $W$. We divide the learning rate by 2 every time our unsupervised validation criterion decreases.

### 3.2 DISCRIMINATOR INPUTS

The embedding quality of rare words is generally not as good as the one of frequent words (Luong et al., 2013), and we observed that feeding the discriminator with rare words had a small, but not negligible negative impact. As a result, we only feed the discriminator with the 50,000 most frequent words. At each training step, the word embeddings given to the discriminator are sampled uniformly. Sampling them according to the word frequency did not have any noticeable impact on the results.

### 3.3 ORTHOGONALITY

Smith et al. (2017) showed that imposing an orthogonal constraint to the linear operator led to better performance. Using an orthogonal matrix has several advantages. First, it ensures that the monolingual quality of the embeddings is preserved. Indeed, an orthogonal matrix preserves the dot product of vectors, as well as their $\ell_2$ distances, and is therefore an isometry of the Euclidean space (such as a rotation). Moreover, it made the training procedure more stable in our experiments. In this work, we propose to use a simple update step to ensure that the matrix $W$ stays close to an orthogonal matrix during training (Cisse et al. (2017)). Specifically, we alternate the update of our model with the following update rule on the matrix $W$:

$$W \leftarrow (1 + \beta)W - \beta(WW^T)W \tag{7}$$

where $\beta = 0.01$ is usually found to perform well. This method ensures that the matrix stays close to the manifold of orthogonal matrices after each update. In practice, we observe that the eigenvalues of our matrices all have a modulus close to 1, as expected.

### 3.4 DICTIONARY GENERATION

The refinement step requires to generate a new dictionary at each iteration. In order for the Procrustes solution to work well, it is best to apply it on correct word pairs. As a result, we use the CSLS method described in Section 2.3 to select more accurate translation pairs in the dictionary. To increase even more the quality of the dictionary, and ensure that $W$ is learned from correct translation pairs, we only consider mutual nearest neighbors, i.e. pairs of words that are mutually nearest neighbors of each other according to CSLS. This significantly decreases the size of the generated dictionary, but improves its accuracy, as well as the overall performance.

### 3.5 VALIDATION CRITERION FOR UNSUPERVISED MODEL SELECTION

Selecting the best model is a challenging, yet important task in the unsupervised setting, as it is not possible to use a validation set (using a validation set would mean that we possess parallel data). To

---

[2]Word vectors downloaded from: `https://github.com/facebookresearch/fastText`

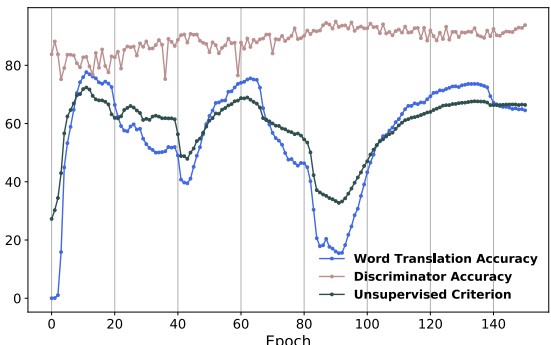

**Figure 2: Unsupervised model selection.** Correlation between our unsupervised validation criterion (black line) and actual word translation accuracy (blue line). In this particular experiment, the selected model is at epoch 10. Observe how our criterion is well correlated with translation accuracy.

address this issue, we perform model selection using an unsupervised criterion that quantifies the closeness of the source and target embedding spaces. Specifically, we consider the 10k most frequent source words, and use CSLS to generate a translation for each of them. We then compute the average cosine similarity between these deemed translations, and use this average as a validation metric. We found that this simple criterion is better correlated with the performance on the evaluation tasks than optimal transport distances such as the Wasserstein distance (Rubner et al. (2000)). Figure 2 shows the correlation between the evaluation score and this unsupervised criterion (without stabilization by learning rate shrinkage). We use it as a stopping criterion during training, and also for hyper-parameter selection in all our experiments.

# 4 EXPERIMENTS

In this section, we empirically demonstrate the effectiveness of our unsupervised approach on several benchmarks, and compare it with state-of-the-art supervised methods. We first present the cross-lingual evaluation tasks that we consider to evaluate the quality of our cross-lingual word embeddings. Then, we present our baseline model. Last, we compare our unsupervised approach to our baseline and to previous methods. In the appendix, we offer a complementary analysis on the alignment of several sets of English embeddings trained with different methods and corpora.

## 4.1 EVALUATION TASKS

**Word translation** The task considers the problem of retrieving the translation of given source words. The problem with most available bilingual dictionaries is that they are generated using online tools like Google Translate, and do not take into account the polysemy of words. Failing to capture word polysemy in the vocabulary leads to a wrong evaluation of the quality of the word embedding space. Other dictionaries are generated using phrase tables of machine translation systems, but they are very noisy or trained on relatively small parallel corpora. For this task, we create high-quality

| | en-es | es-en | en-fr | fr-en | en-de | de-en | en-ru | ru-en | en-zh | zh-en | en-eo | eo-en |
|---|---|---|---|---|---|---|---|---|---|---|---|---|
| *Methods with cross-lingual supervision and fastText embeddings* | | | | | | | | | | | | |
| Procrustes - NN | 77.4 | 77.3 | 74.9 | 76.1 | 68.4 | 67.7 | 47.0 | 58.2 | 40.6 | 30.2 | 22.1 | 20.4 |
| Procrustes - ISF | 81.1 | 82.6 | 81.1 | 81.3 | 71.1 | 71.5 | 49.5 | 63.8 | 35.7 | **37.5** | 29.0 | 27.9 |
| Procrustes - CSLS | 81.4 | 82.9 | 81.1 | **82.4** | 73.5 | **72.4** | **51.7** | **63.7** | **42.7** | 36.7 | **29.3** | 25.3 |
| *Methods without cross-lingual supervision and fastText embeddings* | | | | | | | | | | | | |
| Adv - NN | 69.8 | 71.3 | 70.4 | 61.9 | 63.1 | 59.6 | 29.1 | 41.5 | 18.5 | 22.3 | 13.5 | 12.1 |
| Adv - CSLS | 75.7 | 79.7 | 77.8 | 71.2 | 70.1 | 66.4 | 37.2 | 48.1 | 23.4 | 28.3 | 18.6 | 16.6 |
| Adv - Refine - NN | 79.1 | 78.1 | 78.1 | 78.2 | 71.3 | 69.6 | 37.3 | 54.3 | 30.9 | 21.9 | 20.7 | 20.6 |
| Adv - Refine - CSLS | **81.7** | **83.3** | **82.3** | 82.1 | **74.0** | 72.2 | 44.0 | 59.1 | 32.5 | 31.4 | 28.2 | **25.6** |

**Table 1: Word translation retrieval P@1 for our released vocabularies in various language pairs.** We consider 1,500 source test queries, and 200k target words for each language pair. We use fastText embeddings trained on Wikipedia. NN: nearest neighbors. ISF: inverted softmax. ('en' is English, 'fr' is French, 'de' is German, 'ru' is Russian, 'zh' is classical Chinese and 'eo' is Esperanto)

| | English to Italian | | | Italian to English | | |
|---|---|---|---|---|---|---|
| | P@1 | P@5 | P@10 | P@1 | P@5 | P@10 |
| *Methods with cross-lingual supervision (WaCky)* | | | | | | |
| Mikolov et al. (2013b) [†] | 33.8 | 48.3 | 53.9 | 24.9 | 41.0 | 47.4 |
| Dinu et al. (2015)[†] | 38.5 | 56.4 | 63.9 | 24.6 | 45.4 | 54.1 |
| CCA[†] | 36.1 | 52.7 | 58.1 | 31.0 | 49.9 | 57.0 |
| Artetxe et al. (2017) | 39.7 | 54.7 | 60.5 | 33.8 | 52.4 | 59.1 |
| Smith et al. (2017)[†] | 43.1 | 60.7 | 66.4 | 38.0 | 58.5 | 63.6 |
| Procrustes - CSLS | 44.9 | 61.8 | 66.6 | 38.5 | 57.2 | 63.0 |
| *Methods without cross-lingual supervision (WaCky)* | | | | | | |
| Adv - Refine - CSLS | 45.1 | 60.7 | 65.1 | 38.3 | 57.8 | 62.8 |
| *Methods with cross-lingual supervision (Wiki)* | | | | | | |
| Procrustes - CSLS | 63.7 | 78.6 | 81.1 | 56.3 | 76.2 | 80.6 |
| *Methods without cross-lingual supervision (Wiki)* | | | | | | |
| Adv - Refine - CSLS | **66.2** | **80.4** | **83.4** | **58.7** | **76.5** | **80.9** |

**Table 2: English-Italian word translation** average precisions (@1, @5, @10) from 1.5k source word queries using 200k target words. Results marked with the symbol [†] are from Smith et al. (2017). Wiki means the embeddings were trained on Wikipedia using fastText. Note that the method used by Artetxe et al. (2017) does not use the same supervision as other supervised methods, as they only use numbers in their initial parallel dictionary.

dictionaries of up to 100k pairs of words using an internal translation tool to alleviate this issue. We make these dictionaries publicly available as part of the MUSE library[3].

We report results on these bilingual dictionaries, as well on those released by Dinu et al. (2015) to allow for a direct comparison with previous approaches. For each language pair, we consider 1,500 query source and 200k target words. Following standard practice, we measure how many times one of the correct translations of a source word is retrieved, and report precision@$k$ for $k = 1, 5, 10$.

**Cross-lingual semantic word similarity** We also evaluate the quality of our cross-lingual word embeddings space using word similarity tasks. This task aims at evaluating how well the cosine similarity between two words of different languages correlates with a human-labeled score. We use the SemEval 2017 competition data (Camacho-Collados et al. (2017)) which provides large, high-quality and well-balanced datasets composed of nominal pairs that are manually scored according to a well-defined similarity scale. We report Pearson correlation.

**Sentence translation retrieval** Going from the word to the sentence level, we consider bag-of-words aggregation methods to perform sentence retrieval on the Europarl corpus. We consider 2,000 source sentence queries and 200k target sentences for each language pair and report the precision@$k$ for $k = 1, 5, 10$, which accounts for the fraction of pairs for which the correct translation of the source words is in the $k$-th nearest neighbors. We use the idf-weighted average to merge word into sentence embeddings. The idf weights are obtained using other 300k sentences from Europarl.

## 4.2 RESULTS AND DISCUSSION

In what follows, we present the results on word translation retrieval using our bilingual dictionaries in Table 1 and our comparison to previous work in Table 2 where we significantly outperform previous approaches. We also present results on the sentence translation retrieval task in Table 3 and the cross-lingual word similarity task in Table 4. Finally, we present results on word-by-word translation for English-Esperanto in Table 5.

**Baselines** In our experiments, we consider a supervised baseline that uses the solution of the Procrustes formula given in (2), and trained on a dictionary of 5,000 source words. This baseline can be combined with different similarity measures: NN for nearest neighbor similarity, ISF for Inverted SoftMax and the CSLS approach described in Section 2.2.

**Cross-domain similarity local scaling** This approach has a single parameter $K$ defining the size of the neighborhood. The performance is very stable and therefore $K$ does not need cross-validation: the results are essentially the same for $K = 5, 10$ and $50$, therefore we set $K = 10$ in all experiments. In Table 1, we observe the impact of the similarity metric with the Procrustes supervised approach. Looking at the difference between *Procrustes-NN* and *Procrustes-CSLS*, one can see that CSLS

---

[3]https://github.com/facebookresearch/MUSE

|  | English to Italian | | | Italian to English | | |
|---|---|---|---|---|---|---|
|  | P@1 | P@5 | P@10 | P@1 | P@5 | P@10 |
| *Methods with cross-lingual supervision* | | | | | | |
| Mikolov et al. (2013b) [†] | 10.5 | 18.7 | 22.8 | 12.0 | 22.1 | 26.7 |
| Dinu et al. (2015) [†] | 45.3 | 72.4 | 80.7 | 48.9 | 71.3 | 78.3 |
| Smith et al. (2017) [†] | 54.6 | 72.7 | 78.2 | 42.9 | 62.2 | 69.2 |
| Procrustes - NN | 42.6 | 54.7 | 59.0 | 53.5 | 65.5 | 69.5 |
| Procrustes - CSLS | **66.1** | 77.1 | 80.7 | **69.5** | **79.6** | **83.5** |
| *Methods without cross-lingual supervision* | | | | | | |
| Adv - CSLS | 42.5 | 57.6 | 63.6 | 47.0 | 62.1 | 67.8 |
| Adv - Refine - CSLS | 65.9 | **79.7** | **83.1** | 69.0 | **79.7** | 83.1 |

Table 3: **English-Italian sentence translation retrieval**. We report the average P@k from 2,000 source queries using 200,000 target sentences. We use the same embeddings as in Smith et al. (2017). Their results are marked with the symbol [†].

provides a strong and robust gain in performance across all language pairs, with up to 7.2% in en-eo. We observe that *Procrustes-CSLS* is almost systematically better than *Procrustes-ISF*, while being computationally faster and not requiring hyper-parameter tuning. In Table 2, we compare our *Procrustes-CSLS* approach to previous models presented in Mikolov et al. (2013b); Dinu et al. (2015); Smith et al. (2017); Artetxe et al. (2017) on the English-Italian word translation task, on which state-of-the-art models have been already compared. We show that our *Procrustes-CSLS* approach obtains an accuracy of 44.9%, outperforming all previous approaches. In Table 3, we also obtain a strong gain in accuracy in the Italian-English sentence retrieval task using CSLS, from 53.5% to 69.5%, outperforming previous approaches by an absolute gain of more than 20%.

**Impact of the monolingual embeddings** For the word translation task, we obtained a significant boost in performance when considering fastText embeddings trained on Wikipedia, as opposed to previously used CBOW embeddings trained on the WaCky datasets (Baroni et al. (2009)), as can been seen in Table 2. Among the two factors of variation, we noticed that this boost in performance was mostly due to the change in corpora. The fastText embeddings, which incorporates more syntactic information about the words, obtained only two percent more accuracy compared to CBOW embeddings trained on the same corpus, out of the 18.8% gain. We hypothesize that this gain is due to the similar co-occurrence statistics of Wikipedia corpora. Figure 3 in the appendix shows results on the alignment of different monolingual embeddings and concurs with this hypothesis. We also obtained better results for monolingual evaluation tasks such as word similarities and word analogies when training our embeddings on the Wikipedia corpora.

**Adversarial approach** Table 1 shows that the adversarial approach provides a strong system for learning cross-lingual embeddings without parallel data. On the es-en and en-fr language pairs, *Adv-CSLS* obtains a P@1 of 79.7% and 77.8%, which is only 3.2% and 3.3% below the supervised approach. Additionally, we observe that most systems still obtain decent results on distant languages that do not share a common alphabet (en-ru and en-zh), for which method exploiting identical character strings are just not applicable (Artetxe et al. (2017)). This method allows us to build a strong synthetic vocabulary using similarities obtained with CSLS. The gain in absolute accuracy observed with CSLS on the Procrustes method is even more important here, with differences between *Adv-NN* and *Adv-CSLS* of up to 8.4% on es-en. As a simple baseline, we tried to match the first two moments of the projected source and target embeddings, which amounts to solving $W^{\star} \in \operatorname{argmin}_W \|(WX)^T(WX) - Y^TY\|_{\mathrm{F}}$ and solving the sign ambiguity (Umeyama, 1988). This attempt was not successful, which we explain by the fact that this method tries to align only the first two moments, while adversarial training matches all the moments and can learn to focus on specific areas of the distributions instead of considering global statistics.

**Refinement: closing the gap with supervised approaches** The refinement step on the synthetic bilingual vocabulary constructed after adversarial training brings an additional and significant gain in performance, closing the gap between our approach and the supervised baseline. In Table 1, we observe that our unsupervised method even outperforms our strong supervised baseline on en-it and en-es, and is able to retrieve the correct translation of a source word with up to 83% accuracy. The better performance of the unsupervised approach can be explained by the strong similarity of co-occurrence statistics between the languages, and by the limitation in the supervised approach that uses a pre-defined fixed-size vocabulary (of 5,000 unique source words): in our case the refinement step can potentially use more anchor points. In Table 3, we also observe a strong gain in accuracy

| SemEval 2017 | en-es | en-de | en-it |
|---|---|---|---|
| *Methods with cross-lingual supervision* | | | |
| NASARI | 0.64 | 0.60 | 0.65 |
| our baseline | 0.72 | 0.72 | 0.71 |
| *Methods without cross-lingual supervision* | | | |
| Adv | 0.69 | 0.70 | 0.67 |
| Adv - Refine | 0.71 | 0.71 | 0.71 |

**Table 4: Cross-lingual wordsim task.** NASARI (Camacho-Collados et al. (2016)) refers to the official SemEval2017 baseline. We report Pearson correlation.

| | en-eo | eo-en |
|---|---|---|
| Dictionary - NN | 6.1 | 11.9 |
| Dictionary - CSLS | 11.1 | 14.3 |

**Table 5: BLEU score on English-Esperanto.** Although being a naive approach, word-by-word translation is enough to get a rough idea of the input sentence. The quality of the generated dictionary has a significant impact on the BLEU score.

(up to 15%) on sentence retrieval using bag-of-words embeddings, which is consistent with the gain observed on the word retrieval task.

**Application to a low-resource language pair and to machine translation** Our method is particularly suited for low-resource languages for which there only exists a very limited amount of parallel data. We apply it to the English-Esperanto language pair. We use the fastText embeddings trained on Wikipedia, and create a dictionary based on an online lexicon. The performance of our unsupervised approach on English-Esperanto is of 28.2%, compared to 29.3% with the supervised method. On Esperanto-English, our unsupervised approach obtains 25.6%, which is 1.3% better than the supervised method. The dictionary we use for that language pair does not take into account the polysemy of words, which explains why the results are lower than on other language pairs. People commonly report the P@5 to alleviate this issue. In particular, the P@5 for English-Esperanto and Esperanto-English is of 46.5% and 43.9% respectively.

To show the impact of such a dictionary on machine translation, we apply it to the English-Esperanto Tatoeba corpora (Tiedemann, 2012). We remove all pairs containing sentences with unknown words, resulting in about 60k pairs. Then, we translate sentences in both directions by doing word-by-word translation. In Table 5, we report the BLEU score with this method, when using a dictionary generated using nearest neighbors, and CSLS. With CSLS, this naive approach obtains 11.1 and 14.3 BLEU on English-Esperanto and Esperanto-English respectively. Table 6 in the appendix shows some examples of sentences in Esperanto translated into English using word-by-word translation. As one can see, the meaning is mostly conveyed in the translated sentences, but the translations contain some simple errors. For instance, the "mi" is translated into "sorry" instead of "i", etc. The translations could easily be improved using a language model.

## 5 RELATED WORK

Work on *bilingual lexicon induction* without parallel corpora has a long tradition, starting with the seminal works by Rapp (1995) and Fung (1995). Similar to our approach, they exploit the Harris (1954) distributional structure, but using discrete word representations such as TF-IDF vectors. Following studies by Fung & Yee (1998); Rapp (1999); Schafer & Yarowsky (2002); Koehn & Knight (2002); Haghighi et al. (2008); Irvine & Callison-Burch (2013) leverage statistical similarities between two languages to learn small dictionaries of a few hundred words. These methods need to be initialized with a seed bilingual lexicon, using for instance the edit distance between source and target words. This can be seen as prior knowledge, only available for closely related languages. There is also a large amount of studies in statistical decipherment, where the machine translation problem is reduced to a deciphering problem, and the source language is considered as a ciphertext (Ravi & Knight, 2011; Pourdamghani & Knight, 2017). Although initially not based on distributional semantics, recent studies show that the use of word embeddings can bring significant improvement in statistical decipherment (Dou et al., 2015).

The rise of distributed word embeddings has revived some of these approaches, now with the goal of aligning embedding spaces instead of just aligning vocabularies. Cross-lingual word embeddings can be used to extract bilingual lexicons by computing the nearest neighbor of a source word, but also allow other applications such as sentence retrieval or cross-lingual document classification (Klementiev et al., 2012). In general, they are used as building blocks for various cross-lingual language processing systems. More recently, several approaches have been proposed to learn bilingual dictionaries mapping from the source to the target space (Mikolov et al., 2013b; Zou et al., 2013; Faruqui

& Dyer, 2014; Ammar et al., 2016). In particular, Xing et al. (2015) showed that adding an orthogonality constraint to the mapping can significantly improve performance, and has a closed-form solution. This approach was further referred to as the Procrustes approach in Smith et al. (2017).

The hubness problem for cross-lingual word embedding spaces was investigated by Dinu et al. (2015). The authors added a correction to the word retrieval algorithm by incorporating a nearest neighbors reciprocity term. More similar to our cross-domain similarity local scaling approach, Smith et al. (2017) introduced the inverted-softmax to down-weight similarities involving often-retrieved hub words. Intuitively, given a query source word and a candidate target word, they estimate the probability that the candidate translates back to the query, rather than the probability that the query translates to the candidate.

Recent work by Smith et al. (2017) leveraged identical character strings in both source and target languages to create a dictionary with low supervision, on which they applied the Procrustes algorithm. Similar to this approach, recent work by Artetxe et al. (2017) used identical digits and numbers to form an initial seed dictionary, and performed an update similar to our refinement step, but iteratively until convergence. While they showed they could obtain good results using as little as twenty parallel words, their method still needs cross-lingual information and is not suitable for languages that do not share a common alphabet. For instance, the method of Artetxe et al. (2017) on our dataset does not work on the word translation task for any of the language pairs, because the digits were filtered out from the datasets used to train the fastText embeddings. This iterative EM-based algorithm initialized with a seed lexicon has also been explored in other studies (Haghighi et al., 2008; Kondrak et al., 2017).

There has been a few attempts to align monolingual word vector spaces with no supervision. Similar to our work, Zhang et al. (2017b) employed adversarial training, but their approach is different than ours in multiple ways. First, they rely on sharp drops of the discriminator accuracy for model selection. In our experiments, their model selection criterion does not correlate with the overall model performance, as shown in Figure 2. Furthermore, it does not allow for hyper-parameters tuning, since it selects the best model over a single experiment. We argue it is a serious limitation, since the best hyper-parameters vary significantly across language pairs. Despite considering small vocabularies of a few thousand words, their method obtained weak results compared to supervised approaches. More recently, Zhang et al. (2017a) proposed to minimize the earth-mover distance after adversarial training. They compare their results only to their supervised baseline trained with a small seed lexicon, which is one to two orders of magnitude smaller than what we report here.

## 6 Conclusion

In this work, we show for the first time that one can align word embedding spaces without any cross-lingual supervision, *i.e.*, solely based on unaligned datasets of each language, while reaching or outperforming the quality of previous supervised approaches in several cases. Using adversarial training, we are able to initialize a linear mapping between a source and a target space, which we also use to produce a synthetic parallel dictionary. It is then possible to apply the same techniques proposed for supervised techniques, namely a Procrustean optimization. Two key ingredients contribute to the success of our approach: First we propose a simple criterion that is used as an effective unsupervised validation metric. Second we propose the similarity measure CSLS, which mitigates the hubness problem and drastically increases the word translation accuracy. As a result, our approach produces high-quality dictionaries between different pairs of languages, with up to 83.3% on the Spanish-English word translation task. This performance is on par with supervised approaches. Our method is also effective on the English-Esperanto pair, thereby showing that it works for low-resource language pairs, and can be used as a first step towards unsupervised machine translation.

### Acknowledgments

We thank Juan Miguel Pino, Moustapha Cissé, Nicolas Usunier, Yann Ollivier, David Lopez-Paz, Alexandre Sablayrolles, and the FAIR team for useful comments and discussions.

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

## 7 APPENDIX

In order to gain a better understanding of the impact of using similar corpora or similar word embedding methods, we investigated merging two English monolingual embedding spaces using either Wikipedia or the Gigaword corpus (Parker et al. (2011)), and either Skip-Gram, CBOW or fastText methods (see Figure 3).

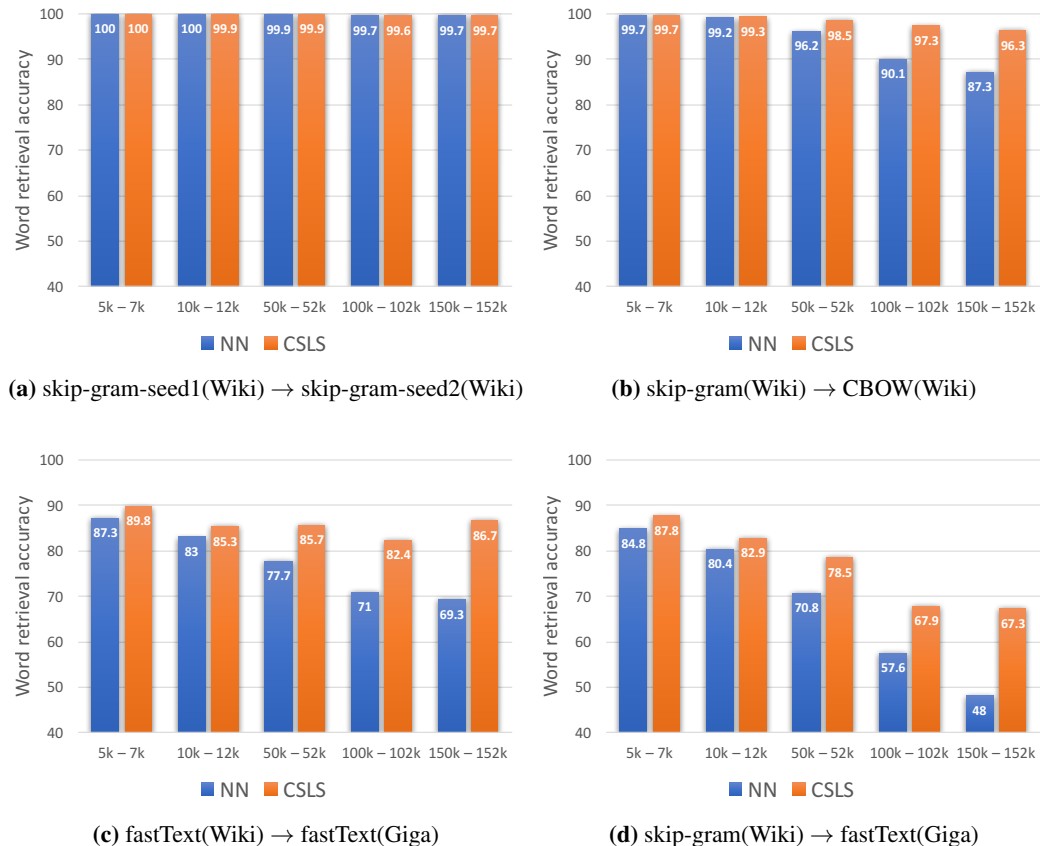

(a) skip-gram-seed1(Wiki) → skip-gram-seed2(Wiki)

(b) skip-gram(Wiki) → CBOW(Wiki)

(c) fastText(Wiki) → fastText(Giga)

(d) skip-gram(Wiki) → fastText(Giga)

**Figure 3: English to English word alignment accuracy.** Evolution of word translation retrieval accuracy with regard to word frequency, using either Wikipedia (Wiki) or the Gigaword corpus (Giga), and either skip-gram, continuous bag-of-words (CBOW) or fastText embeddings. The model can learn to perfectly align embeddings trained on the same corpus but with different seeds (a), as well as embeddings learned using different models (overall, when employing CSLS which is more accurate on rare words) (b). However, the model has more trouble aligning embeddings trained on different corpora (Wikipedia and Gigaword) (c). This can be explained by the difference in co-occurrence statistics of the two corpora, particularly on the rarer words. Performance can be further deteriorated by using both different models and different types of corpus (d).

| Source | mi kelkfoje parolas kun mia najbaro tra la barilo . |
|---|---|
| Hypothesis | sorry sometimes speaks with my neighbor across the barrier . |
| Reference | i sometimes talk to my neighbor across the fence . |
| Source | la viro malanta ili ludas la pianon . |
| Hypothesis | the man behind they plays the piano . |
| Reference | the man behind them is playing the piano . |
| Source | bonvole protektu min kontra tiuj malbonaj viroj . |
| Hypothesis | gratefully protects hi against those worst men . |
| Reference | please defend me from such bad men . |

**Table 6: Esperanto-English.** Examples of fully unsupervised word-by-word translations. The translations reflect the meaning of the source sentences, and could potentially be improved using a simple language model.

