# OpenReview forum: "Word translation without parallel data"
_ICLR.cc/2018/Conference — Accept (Poster)_

### Official Review · AnonReviewer1 · 2017-11-20
**Good and interesting work (a few issues on the paper's claims)**

**Rating:** 9
**Confidence:** 4

**Review:**

This paper presents a new method for obtaining a bilingual dictionary, without requiring any parallel data between the source and target languages. The method consists of an adversarial approach for aligning two monolingual word embedding spaces, followed by a refinement step using frequent aligned words (according to the adversarial mapping). The approach is evaluated on single word translation, cross-lingual word similarity, and sentence translation retrieval tasks.

The paper presents an interesting approach which achieves good performance. The work is presented clearly, the approach is well-motivated and related to previous studies, and a thorough evaluation is performed.

My one concern is that the supervised approach that the paper compares to is limited: it is trained on a small fixed number of anchor points, while the unsupervised method uses significantly more words. I think the paper's comparisons are valid, but the abstract and introduction make very strong claims about outperforming "state-of-the-art supervised approaches". I think either a stronger supervised baseline should be included (trained on comparable data as the unsupervised approach), or the language/claims in the paper should be softened. The same holds for statements like "... our method is a first step ...", which is very hard to justify. I also do not think it is necessary to over-sell, given the solid work in the paper.

Further comments, questions and suggestions:
- It might be useful to add more details of your actual approach in the Abstract, not just what it achieves.
- Given you use trained word embeddings, it is not a given that the monolingual word embedding spaces would be alignable in a linear way. The actual word embedding method, therefore, has a big influence on performance (as you show). Could you comment on how crucial it would be to train monolingual embedding spaces on similar domains/data with similar co-occurrence statistics, in order for your method to be appropriate?
- Would it be possible to add weights to the terms in eq. (6), or is this done implicitly?
- How were the 5k source words for Procrustes supervised baseline selected?
- Have you considered non-linear mappings, or jointly training the monolingual word embeddings while attempting the linear mapping between embedding spaces?
- Do you think your approach would benefit from having a few parallel training points?

Some minor grammatical mistakes/typos (nitpicking):
- "gives a good performance" -> "gives good performance"
- "Recent works", "several works", "most works", etc. -> "recent studies", "several studies", etc.
- "i.e, the improvements" -> "i.e., the improvements"

The paper is well-written, relevant and interesting. I therefore recommend that the paper be accepted.

---

> ### Author Response · Authors · 2017-12-28
> **response 1**
>
> We thank the reviewer for the feedback and comments.
>
> It is true that the supervised approach is limited in the sense that it only considers 5000 pairs of words. However, previous works have shown that using more than 5000 pairs of words does not improve the performance (Artetxe et al. (2017)), and can even be detrimental (see Dinu et al. (2015)). This is why we decided to consider 5000 pairs only, to be consistent with previous works. Also, note that we made our supervised baseline (Procrustes + CSLS) as strong as possible, and it is actually state-of-the-art.
>
> Regarding the claim "this is a first step towards fully unsupervised machine translation", what we meant is that the method proposed in the paper could potentially be used in a more complex framework for unsupervised MT at the sentence level. We rephrased this sentence in the updated version of the paper.
>
> We now address the comments / suggestions of the reviewer:
>
> - The abstract could indeed benefit from details about the model. We will add some.
> - The co-occurrence statistics have indeed an impact on the overall performance of the model. This impact is consistent for both supervised and unsupervised approaches. Indeed, our unsupervised method obtains 66.2% accuracy on the English-Italian pair on the Wikipedia corpora (Table 2), and 45.1% accuracy on the UKWAC / ITWAC non-comparable corpora. This result was not in the paper (we thought it was redundant with Table 1), but we added it in Table 2 in the updated version. Figure 3 in the appendix also gives insights about the impact of the similarity of the two domains, by comparing the quality of English-English alignment using embeddings trained on different English corpora.
> - It would indeed possible to add weights in Equation (6). We tried to weight the r_S and r_T terms, but we did not observe a significant improvement compared to the current equation.
> - In the supervised approach, we generated translations for all words from the source language to the target language, and vice-versa (a translation being a pair (x, y) associated with the probability for y of being the correct translation of x). Then, we considered all pairs of words (x, y) such that y has a high probability of being a translation of x, but also that x has a high probability of being a translation of y. Then, we sorted all generated translation pairs by frequency of the source word, and took the 5000 first resulting pairs.
> - We tried to use non-linear mappings (namely a feedforward network with 1 or 2 hidden layers), but in these experiments, the adversarial training was quite unstable, and like in Mikolov et al. (2013), we did not observe better results compared to the linear mapping. Actually, the linear mapping was working significantly better, and since the Procrustes algorithm in the refinement step requires the mapping to be linear, we decided to focus on this type of mapping. Moreover, the linear mapping is convenient because we can impose the orthogonality constraint, which guarantees that the quality of the source monolingual embeddings is preserved after mapping.
> - We did not try to jointly learn the embeddings as well as the mapping, but this is a nice idea and definitely something that needs to be investigated. We think that the joint learning could improve the cross-lingual embeddings, but especially, it could significantly improve the quality of monolingual embeddings on low-resource languages.
> - Our approach would definitely benefit from having a few parallel training points. These points could be used to pretrain the linear mapping for the adversarial training, or even as a validation dataset. This will be the focus of future work.

---

> > ### Comment · AnonReviewer1 · 2018-01-08
> > **Reviewer response**
> >
> > Thank you for the very detailed response, also to the other reviewers' comments: all the questions and concerns were addressed very well.

---

### Official Review · AnonReviewer2 · 2017-11-27

**Rating:** 3
**Confidence:** 5

**Review:**

The paper proposes a method to learn bilingual dictionaries without parallel data using an adversarial technique. The task is interesting and relevant, especially for in low-resource language pair settings.

The paper, however, misses comparison against important work from the literature that is very relevant to their task — decipherment (Ravi, 2013; Nuhn et al., 2012; Ravi & Knight, 2011) and other approaches like CCA.

The former set of works, while focused on machine translation also learns a translation table in the process. Besides, the authors also claim that their approach is particularly suited for low-resource MT and list this as one of their contributions. Previous works have used non-parallel and comparable corpora to learn MT models and for bilingual lexicon induction. The authors seem aware of corpora used in previous works (Tiedemann, 2012) yet provide no comparison against any of these methods. While some of the bilingual lexicon extraction works are cited (Haghighi et al., 2008; Artetxe et al., 2017), they do not demonstrate how their approach performs against these baseline methods. Such a comparison, even on language pairs which share some similarities (e.g., orthography), is warranted to determine the effectiveness of the proposed approach.

The proposed methodology is not novel, it rehashes existing adversarial techniques instead of other probabilistic models used in earlier works.

For the translation task, it would be useful to see performance of a supervised MT baseline (many tools available in open-source) that was trained on similar amount of parallel training data (60k pairs) and see the gap in performance with the proposed approach.

The paper mentions that the approach is “unsupervised”. However, it relies on bootstrapping from word embeddings learned on Wikipedia corpus, which is a comparable corpus even though individual sentences are not aligned across languages. How does the quality degrade if word embeddings had to be learned from scratch or initialized from a different source?

---

> ### Author Response · Authors · 2017-12-28
> **response 2**
>
> We thank the reviewer for the feedback and comments.
>
> The main concern of the review is about the lack of comparisons with existing works.
> - The reviewer reproaches the lack of comparison against CCA, while the comparison against CCA is provided in Table 2. The reviewer also points out the lack of comparison against Artetxe et al. (2017). This comparison is also provided in the paper.
> - We agree that our method could be compared to decipherment techniques, and would have been happy to try the method of Ravi & Knight but there is no open-source version of their code available online (like for Faruqui & Dyer, Dinu et al, Artexte et al, Smith et al). Therefore, considering the large body of literature in that domain, we focused on comparing our approach with the most recent state-of-the-art and supervised approaches, which in our opinion is a fair way to evaluate against reproducible baselines.
>
> The second reviewer concern is about the performance of the model on non-comparable corpora. We considered that this was redundant with the results on Wikipedia provided in Table 1 and Table 2. As explained in one previous comment, our strategy was to first show that our supervised method (Procrustes-CSLS) is state-of-the-art, and then to compare our unsupervised approach against this new baseline. We added the result of our unsupervised approach (Adv - Refine - CSLS) on non-comparable WaCky corpora in Table 2. In particular, our unsupervised model on the non-comparable WaCky datasets is also state of the art with 45.1% accuracy.
>
> The reviewer criticises the lack of novelty. To the best of our knowledge, the fact that an adversarial approach obtains state-of-the-art cross-lingual embeddings is new. Most importantly, the contributions of our paper are not limited to the adversarial approach. The CSLS method introduced to mitigate the hubness problem is new, and improves the state-of-the-art by up to 24% on the sentence retrieval task, as well as it improves the supervised baseline. We also introduced an unsupervised criterion that is highly correlated with the cross-lingual embeddings quality, which is also novel as far as we know, and a key element for training.
>
> Al last, please consider that we made our code publicly available and provided high-quality dictionaries for 110 oriented language pairs to help the community, as this type of resources are very difficult to find online.

---

### Official Review · AnonReviewer3 · 2017-11-28
**Well-rounded contribution, nice read, incomplete related work**

**Rating:** 8
**Confidence:** 3

**Review:**

An unsupervised approach is proposed to build bilingual dictionaries without parallel corpora, by aligning the monolingual word embeddings spaces, i.a. via adversarial learning.

The paper is very well-written and makes for a rather pleasant read, save for some need for down-toning the claims to novelty as voiced in the comment re: Ravi & Knight (2011) or simply in general: it's a very nice paper, I enjoy reading it *in spite*, and not *because* of the text sales-pitching itself at times.

There are some gaps in the awareness of the related work in the sub-field of bilingual lexicon induction, e.g. the work by Vulic & Moens (2016).

The evaluation is for the most part intrinsic, and it would be nice to see the approach applied downstream beyond the simplistic task of English-Esperanto translation: plenty of outlets out there for applying multilingual word embeddings. Would be nice to see at least some instead of the plethora of intrinsic evaluations of limited general interest.

In my view, to conclude, this is still a very nice paper, so I vote clear accept, in hope to see these minor flaws filtered out in the revision.

---

> ### Author Response · Authors · 2017-12-28
> **response 3**
>
> We thank the reviewer for the feedback and comments.
>
> As mentioned in the comments, we added to the paper citations to the work of Ravi & Knight (2011) and some subsequent works on decipherment, and down-toned some claims in the paper.
>
> Thank you for pointing the paper of Vulic & Moens, we were not aware of this paper and we added a citation in the updated version of the paper. Note however that the work of Vulic & Moens relies on document-aligned corpora while our method does not require any form of alignment.
>
> We evaluated the cross-lingual embeddings on 4 different tasks: cross-lingual word similarity, word translation, sentence retrieval, and sentence translation. It is true that the quality of these embeddings on other downstream tasks would be interesting and will be investigated in future work.

---

### Public Comment · (anonymous) · 2017-11-03
**Is it really the first step towards unsupervised MT?**

Saying that the method is a first step towards fully unsupervised machine translation seems like a bold (if not false) statement. In particular, this has been done before using deciphering:

Ravi & Knight, "Deciphering Foreign Language", ACL 2011, http://aclweb.org/anthology/P/P11/P11-1002.pdf

There are plenty of other similar previous work besides this one. I think any claims on MT without parallel corpora should at least mention deciphering as related work.

---

> ### Author Response · Authors · 2017-11-03
> **response**
>
> Thank you for the pointer, we were aware of this work and we will add a citation. Note however that our focus is not to learn to a machine translation system (we just gave a simple example of this application, together with others like sentence retrieval, word similarity, etc.), but to infer a bilingual dictionary without using any labeled data. Unlike Ravi et al. we use monolingual data on both side at training time, and we infer a large bilingual dictionary (200K words). When we say "this is a first step towards fully unsupervised machine translation" it does not mean we are the first to look at this problem, we simply meant that our method could be used as a first step in a more complex pipeline. We will rephrase this sentence to avoid confusion.
> In other words, the two works look at different things: this one is focussed on learning a bilingual dictionary, while the other is focussed on the problem of machine translation.

---

### Public Comment · (anonymous) · 2017-11-27
**Comparison with previous work should be improved**

I think that the paper does not do a good job at comparing the proposed method with previous work.

While most of the experiments are run in a custom dataset and do not include results from previous authors, the paper also reports some results in the standard dataset from Dinu et al. (2015) "to allow for a direct comparison with previous approaches", which I think that is necessary. However, they inexplicably use a different set of embeddings, trained in a different corpus, for their unsupervised method in these experiments, so their results are not actually comparable with the rest of the systems. While I think that these results are also interesting, as they shows that the training corpus and embedding hyperparameters can make a huge difference, I see no reason not to also report the truly comparable results with the standard embeddings used by previous work. In other words, Table 2 is missing a row for "Adv - Refine - CSLS" using the same embeddings as the rest of the systems.

Moreover, I think that the choice of training the embeddings in Wikipedia is somewhat questionable. Wikipedia is a document-aligned comparable corpus, and it seems reasonable that the proposed method could somehow benefit from that, even if it was not originally designed to do so. In other words, while the proposed method is certainly unsupervised in design, I think that it was not tested in truly unsupervised conditions. In fact, there is some previous work that learns cross-lingual word embeddings from Wikipedia by exploiting this document alignment information (http://www.aclweb.org/anthology/P15-1165), which shows that this cross-lingual signal in Wikipedia is actually very strong.

Apart from that, I think that the paper is a bit unfair with some previous work. In particular, the proposed adversarial method is very similar to that of Zhang et al. (2017b), and the authors simply state that the performance of the latter "is significantly below supervised methods", without any experimental evidence that supports this claim. Considering that the implementation of Zhang et al. (2017b) is public (http://nlp.csai.tsinghua.edu.cn/~zm/UBiLexAT/), the authors could have easily tested it in their experiments and show that the proposed method is indeed better than that of Zhang et al. (2017b), but they don't.

I also think that the authors are a bit unfair in their criticism of Artetxe et al. (2017). While the proposed method has the clear advantage of not requiring any cross-lingual signal, not even the assumption of shared numerals in Artetxe et al. (2017), it is not true that the latter is "just not applicable" to "languages that do not share a common alphabet (en-ru and en-zh)", as both Russian and Chinese, as well as many other languages that do not use a latin alphabet, do use arabic numerals. In relation to that, the statement that "the method of Artetxe et al. (2017) on our dataset does not work on the word translation task for any of the language pairs, because the digits were filtered out from the datasets used to train the fastText embeddings" clearly applies to the embeddings they use, and not to the method of Artetxe et al. (2017) itself. Once again, considering that the implementation of Artetxe et al. (2017) is public (https://github.com/artetxem/vecmap), the authors could have easily supported their claims experimentally, but they also fail to do so.

---

> ### Author Response · Authors · 2017-11-27
> **comparison**
>
> The paper by Dinu et al. provides embeddings and dictionaries for the English-Italian language pair. The embeddings they provide have become pretty standard and we found at least 5 previous methods that used this dataset:
> Mikolov et al., Faruqui et al., Dinu et al., Smith et al., Artetxe et al.
> These previous papers provide strong supervised SOTA baselines on the word translation task, and in Table 2 we show results of our supervised method compared to these 5 papers. The row “Procrustes + CSLS” is a supervised baseline, training our method with supervision using exactly the same word embeddings and dictionaries as in Dinu et al. These results show that our supervised baseline works better than all these previous approaches (reaching 44.9% P@1 en-it).
> The requested unsupervised configuration “Adv - Refine - CSLS” using the same embeddings and dictionary as in Dinu et al. obtains 45.1% on en-it, which is better than our supervised baseline (and SOTA by more than 2%).
>
> However, this information is redundant with Table 1, which shows that our unsupervised approach is better than our supervised baseline on European languages. We therefore decided not to incorporate this result, but we will add it back as suggested.
>
> Moreover, using the Wacky datasets (non comparable corpora) to learn embeddings, we improved the SOTA by 11.5% and 26.6% on the sentence retrieval task using our CSLS method, see table 3. Again, these experiments use the very same setting as previously reported in the literature.
>
> More generally, regarding your comment “they inexplicably use a different set of embeddings, trained in a different corpus”, note that:
> - As noted above, we did compare using the very same embeddings and settings as others.
> - We did study the effect of using different corpora: see fig. 3
> - As shown in the paper, using our method on Wikipedia improves the results by more than 20%
> - Wikipedia is available in most languages, pretrained embeddings were already released and publicly available, we just downloaded them (while the Wacky datasets are only available for a few languages)
> - We found that the monolingual quality of these pretrained embeddings is better than the one obtained on the Wacky datasets
>
> As opposed to the 5 methods we compare ourselves against in the paper, Zhang et al. (2017):
> 1) used different embeddings and dictionaries which they do not provide,
> 2) used a lexicon of 50 or 100 word pairs only in their supervised baseline, which is different than standard practice since Mikolov et al. (2013b) (see Dinu et al., Faruqui et al., Smith et al., etc.) and which is also what we did, namely considering dictionaries with 5000 pairs. As a result, they compare themselves to a very weak baseline.
> 3) in the retrieval task they consider a very simplistic settings, with only a few thousands words, as opposed to large dictionaries of 200k words (as done by Dinu et al., Smith et al. and us).
> 4) they. do not provide a validation set and, as shown in Figure 2 in our paper, their stopping criterion does not work well.
> We did try to run their code, but we have not been successful yet.
>
> As for comparing against Artetxe et al., as reported in table 2 they obtain a P@1 of 39.7% while we obtain 45.1% using the same Dinu’s embeddings.
>
> Finally, we have released our code, along with our embeddings / dictionaries for reproducibility. We will share the link here as soon as the decisions are out in order to preserve anonymity.

---

> > ### Public Comment · (anonymous) · 2017-11-28
> > **Answer**
> >
> > I find the answer from the authors quite satisfying. In particular, the missing result that was provided in the answer (and should be definitely added to the paper) addresses my main concerns regarding the comparability with previous work. While this result is not as spectacular as the others (almost at par with the supervised system, possibly because the comparability of Wikipedia is playing a role as pointed in my previous comment) and I think that some of the claims in the paper should be reworded accordingly, it does convincingly show that the proposed method can achieve SOTA results in a standard dataset without any supervision.
> >
> > Regarding the work of Zhang et al. (2017b) and Artetxe et al. (2017), I agree on most of the comments on the former, and this new result shows that the proposed method works better than the latter. However, I still think that some of the claims regarding these papers (e.g. Zhang et al. (2017b) "is significantly below supervised methods" or Artetxe et al. (2017) does not work for en-ru and en-zh) are unfounded and need to either be supported experimentally or reconsidered.

---

### Public Comment · (anonymous) · 2017-12-23
**Methodological Distinction from Zhang et al. not clear**

While the results in this paper are very nice, this method seems to be almost the same as Zhang et al., and even after reading the comments in the discussion I can't tell what the main methodological differences are. Is it really only the stopping criterion for training? If so, the title "word translation without parallel data" seems quite grandiose, and it should probably be something more like "a better stopping criterion for word translation without parallel data".

I'm relatively familiar with this field, and if it's even difficult for me to tell the differences between this and highly relevant previous work I'm worried that it will be even more difficult for others to put this research in the appropriate context.

---

> ### Author Response · Authors · 2017-12-29
> **title**
>
> We thank you for your comment and we are glad to clarify.
>
> The methodological difference between what we have proposed and Zhang et al.’s method is not just a better stopping criterion, but more importantly, a better underlying method. Here is the detailed comparison between the two approaches:
> - The very first step which is adversarial training with orthogonality constraint, is similar, see figure 1B in sec. 2.1 of our paper and figure 2 in [Zhang et al 2017a] (except for the use of earth mover distance) and figure 2b in [Zhang et al 2017b],  but:
> - the refinement step described in sec. 2.2 and Figure 1C is not present is Zhang et al 2017a/b, nor is
> - the use of CSLS metric addressing the hubness problem, see sec 2.3 and figure 1D.
> In contrast, we do not use any of the approaches described in Zhang et al. 2017b shown in their figure 2b and 2c.
> Empirically, we demonstrate in tab. 1 the importance of both the refinement step and the use of CSLS metric to achieve excellent performance.
>
> In addition to this key differences between the two approaches, we have also proposed a better stopping criterion as pointed out in your comment.  This is actually not just a stopping criterion but a “validation” criterion that quantifies the closeness of the source and target spaces, and that correlates well with the word translation accuracy (see Figure 2). We not only use it as a stopping criterion, but also to select the best models across several experiments, which is something that Zhang et al. cannot do. Moreover, their stopping criterion is based on “sharp drops of the generator loss”, and it did not work in our experiments to select the best models (see Figure 2 of our paper).
>
> In terms of evaluation protocol, Zhang et al. compare their unsupervised approach with a supervised method trained on 50 or 100 pairs of words only, which is a little odd given that most papers consider 5000 pairs of words (see Mikolov et al., Dinu et al., Faruqui et al., Smith et al., Artetxe et al., etc.). As a result, they have an extremely weak supervised baseline, while our supervised baseline is itself the new state of the art.
>
> Finally, note that we have released our code and we know that other research groups were able to already reproduce our results, and we have also released our ground-truth dictionaries and evaluation pipeline, which will hopefully help the community make further strides in this area (as today we lack a standardized evaluation protocol as pointed out above, and large scale ground truth dictionaries in lots of different language pairs).

---

### Author Response · Authors · 2017-12-28
**review responses**

We thank all the reviewers for the feedback and comments. We replied to each of them individually, and uploaded a revised version of the paper. In particular, we:
- Rephrased one of the claims made in the abstract about unsupervised machine translation
- Added the requested 45.1% result of our unsupervised approach on the WaCky datasets
- Fixed some typos
- Added missing citations

---

### Decision · Program_Chairs · 2018-01-29
**ICLR 2018 Conference Acceptance Decision**

**Decision:**

Accept (Poster)

**Comment:**

There is significant discussion on this paper and high variance between reviewers:  one reviewer gave the paper a low score.  However the committee feels that this paper should be accepted at the conference since it provides a better framework for reproducibility, performs more large scale experiments than prior work.  One small issue the lack of comparison in terms of empirical results between this work and Zhang et al's work, but the responses provided to both the reviewers and anonymous commenters seem to be satisfactory.